# Selenium Nanoparticles Attenuate Cobalt Nanoparticle-Induced Skeletal Muscle Injury: A Study Based on Myoblasts and Zebrafish

**DOI:** 10.3390/toxics12020130

**Published:** 2024-02-06

**Authors:** Zejiu Tan, Linhua Deng, Zhongjing Jiang, Gang Xiang, Gengming Zhang, Sihan He, Hongqi Zhang, Yunjia Wang

**Affiliations:** 1Department of Spine Surgery and Orthopaedics, Xiangya Hospital, Central South University, Changsha 410008, China; tzj0909@csu.edu.cn (Z.T.); 228111085@csu.edu.cn (L.D.); zhongjing@csu.edu.cn (Z.J.); csuxyxg@csu.edu.cn (G.X.); 218112315@csu.edu.cn (G.Z.); 238102143@csu.edu.cn (S.H.); zhq9996@csu.edu.cn (H.Z.); 2National Clinical Research Center for Geriatric Disorders, Xiangya Hospital, Central South University, Changsha 410008, China

**Keywords:** cobalt nanoparticles, selenium nanoparticles, muscle injury, myoblast, zebrafish model, metal implants

## Abstract

Cobalt alloys have numerous applications, especially as critical components in orthopedic biomedical implants. However, recent investigations have revealed potential hazards associated with the release of nanoparticles from cobalt-based implants during implantation. This can lead to their accumulation and migration within the body, resulting in adverse reactions such as organ toxicity. Despite being a primary interface for cobalt nanoparticle (CoNP) exposure, skeletal muscle lacks comprehensive long-term impact studies. This study evaluated whether selenium nanoparticles (SeNPs) could mitigate CoNP toxicity in muscle cells and zebrafish models. CoNPs dose-dependently reduced C2C12 viability while elevating reactive oxygen species (ROS) and apoptosis. However, low-dose SeNPs attenuated these adverse effects. CoNPs downregulated myogenic genes and α-smooth muscle actin (α-SMA) expression in C2C12 cells; this effect was attenuated by SeNP cotreatment. Zebrafish studies confirmed CoNP toxicity, as it decreased locomotor performance while inducing muscle injury, ROS generation, malformations, and mortality. However, SeNPs alleviated these detrimental effects. Overall, SeNPs mitigated CoNP-mediated cytotoxicity in muscle cells and tissue through antioxidative and antiapoptotic mechanisms. This suggests that SeNP-coated implants could be developed to eliminate cobalt nanoparticle toxicity and enhance the safety of metallic implants.

## 1. Introduction

Metal implants have revolutionized treatment options for many patients in modern medicine. Cobalt alloys, due to their mechanical robustness and biocompatibility, are commonly used in the manufacture of implants, such as artificial joints, bone plates, and screws [1]. The number of applications for cobalt alloys is increasing every year, and millions of tons of cobalt alloys are used each year in the manufacturing of implants [2]. However, as research progresses, it is becoming increasingly clear that cobalt alloy implants release particles of varying sizes as they wear and corrode in vivo [3]. Nanoscale particles are believed to be more toxic [4,5]. These particles can damage the tissues surrounding the implant and may even travel to distant sites through the bloodstream [6]. Since cobalt alloy implants can remain in the body for an individual’s lifetime, there are concerns regarding their potential toxicity [7]. Previous studies have indicated that patients who undergo hip replacement surgery experience significant changes in muscle tissue structure, as well as muscle tissue damage [8]. Furthermore, elevated cobalt concentrations in the blood may suggest the release of cobalt nanoparticles. The authors also observed inflammatory damage in rat skeletal muscle in a model of cobalt-alloyed dorsal muscle implants [9]. Previous research has also reported the gradual accumulation of cobalt in skeletal muscle over time due to the widespread use of cobalt alloys in orthopedics [7]. Therefore, it is necessary to provide a comprehensive description of the toxicity of these metal nanoparticles on various tissues.

Current applications of CoNP in orthopedics include modifying the surface of orthopedic implants to improve their biocompatibility and osseointegration properties. Additionally, CoNP has been used as an orthopedic imaging agent to improve diagnostic accuracy and provide clearer images of bone tissue due to its excellent contrast performance in X-ray and other imaging techniques. The mechanism of toxicity of CoNPs involves multiple pathways. However, research suggests that CoNPs may have adverse effects on the human body due to their ability to penetrate cell membranes and disrupt cellular structure and function. Unlike metallic cobalt and cobalt in other metals, CoNPs have a small size and large surface area, which increases their likelihood of penetrating cell membranes [10,11]. Additionally, CoNPs can modulate the NFκB signaling pathway [12], affecting the inflammatory response [13], and induce the production of reactive oxygen species in cells, triggering oxidative stress and causing cellular damage [14]. Previous studies have reported that exposure of muscle cells to CoNPs or cobalt chloride results in decreased viability and impaired myogenic differentiation [10,15]. This is a significant concern due to the widespread use of cobalt–chromium alloys in orthopedic implants, leading to chronic localized cobalt exposure to muscle tissue. Additionally, the human body may be exposed to CoNPs through various sources, including occupational exposure from industrial production, environmental contamination, and food consumption. However, the effects of CoNP exposure on muscle tissues are not yet clear, making it urgent to assess their toxicity.

Research has consistently shown that metal nanoparticles, such as silver, gold, titanium dioxide, and cobalt, have harmful effects on the viability, differentiation, and tissue regeneration of skeletal muscle cells [16,17,18,19]. At the cellular level, the nanoparticles disrupt key myogenic transcription factors and signaling proteins that control the proliferation and differentiation of muscle progenitor cells into mature myotubes [20]. These factors include MyoD, myogenin, Myf5, and Pax7. Furthermore, the expression of muscle-specific proteins, such as myosin heavy chain, is reduced. Studies in animal models have confirmed the negative effects of CoNPs on muscle regeneration [21]. These effects include smaller regenerating myofibers, incorrect muscle structure, and inflammatory responses. In mice injected with CoNPs, there was a reduction in muscle weight and fiber cross-sectional area, and localized necrosis, as well as inflammatory cell infiltration [10].

Selenium nanoparticles (SeNPs) have emerged as a potential alternative to implantable coatings, as they promise to mitigate the toxicity of metallic biomaterials [22,23]. They exhibit low toxicity, although high doses may still hinder cell proliferation and function [24]. Selenium, a key trace element, plays an important role in cell protection and repair [25]. Additionally, SeNPs have good biocompatibility and excellent bioactivity. SeNPs have been shown to significantly reduce oxidative stress-induced cellular damage and protect cells from further damage by scavenging free radicals and inhibiting inflammatory responses [26]. Additionally, they have been found to promote muscle cell proliferation and repair, which can accelerate the regeneration process of damaged tissues [27,28]. SeNPs stimulate the expression of vascular endothelial growth factor, promoting the formation of new blood vessels [29]. This improves the blood supply to the damaged area, accelerating the tissue repair process. The protective effect of SeNPs against toxic doses of other metal nanoparticles is investigated, providing a theoretical basis for subsequent protective applications of selenium.

This study examines the impact of cobalt and SeNPs on muscle cells at varying concentrations. Specifically, we assessed the effects of low doses of SeNPs on CoNP-induced cell viability, oxidative stress, apoptosis, and myogenic differentiation. Our findings indicate that low concentrations of SeNPs can mitigate the muscle cytotoxicity caused by CoNPs. This study evaluated the protective effects of SeNPs against CoNPs in apoptosis induction, ROS generation, and muscle cell differentiation. Zebrafish experiments verified the muscle-protective effects of SeNPs. The results suggest that SeNPs can alleviate the adverse effects of CoNPs on muscle tissue. The subsequent application of SeNP coatings may be able to counteract the toxicity of cobalt metal implants, promoting the safety of metal implants.

## 2. Materials and Methods

### 2.1. Characterization of Nanoparticles

Cobalt and selenium nanoparticles were obtained from CW-nano Technology Co., Ltd. (Shanghai, China) and Rui Xi Biotechnology Co., Ltd. (Xi’an, China) respectively. Metal nanoparticles are synthesized and manufactured by chemical synthesis according to the protocol of the manufacturer. CoNP and SeNP powders were suspended in sterilized high-purity water at a concentration of 800 mg/µL (stock solution). Prior to each experiment, the NP suspensions were vortexed, placed in precooled glass vials, and sonicated for five 20 s cycles at 65% of the maximum effective power (90 watts) using an ultrasonic device. This was done to minimize particle agglomeration and reduce the effects of high temperatures. The surface morphology of the nanoparticles was observed using a scanning electron microscope (SEM, ZEISS Sigma 300, Ulm, Germany), and their purity was determined through EDAX analysis. The surface charge and hydrodynamic diameter were measured with a Zetasizer Nano ZS90 (Malvern, UK).

### 2.2. Cell Culture

The C2C12 mouse myoblast cell line used in this study was obtained from Procell (Wuhan, China). The cells were cultured in DMEM medium supplemented with 10% fetal bovine serum (FBS) and 1% penicillin–streptomycin under controlled conditions (37 ℃, 5% CO_2_, and 95% humidity). For cell experiments, we prepared stock solutions of CoNPs at concentrations of 1000 μg/mL and 500 μg/mL. To do this, we dispersed Se and CoNPs in ultrapure water and sonicated the mixture for 5 min (50 kJ). Subsequently, the solutions were added to DMEM medium containing 10% FBS. The final concentrations of CoNPs were 0, 1, 5, 10, 20, 40, and 80 μg/mL, while the final concentrations of SeNPs were 0, 0.5, 1, 5, 10, 20, and 40 μg/mL.

### 2.3. Cell Vitality

Cellular vitality was assessed using the CCK-8 method. The cultured cells were incubated in a 96-well plate until they reached 70–80% confluence. Then, the cells were treated with CoNPs at concentrations of 0, 1, 5, 10, 20, 40, and 80 μg/mL for 24 h, and SeNPs at concentrations of 0, 0.5, 1, 5, 10, 20, and 40 μg/mL for 24 h. After exposure to nanoparticles, 10 μL of CCK-8 solution was added to each well, followed by incubation in the cell incubator for 1–4 h. Absorbance at 450 nm was measured using a microplate reader. To ensure reliable data, four sets of replicates were collected for each nanoparticle concentration, and the mean value was subsequently calculated. This study selected CoNP concentrations of 10 and 20 μg/mL based on the CCK-8 experimental results. These concentrations were found to decrease cell viability. Additionally, a SeNP concentration of 5 μg/mL was selected, which significantly promoted cell proliferation. To observe changes in cell viability after combined treatment, solutions containing cobalt and selenium nanoparticles were added simultaneously, and after 24 h, cell viability was assessed using the CCK-8 method.

### 2.4. Confocal Microscope

FITC-conjugated nanoparticles were prepared as follows. Totals of 20 mg each of cobalt and selenium nanoparticles were dissolved in 20 mL of distilled water and sonicated for 15–30 min at 40 Hz. Subsequently, the nanomaterials were introduced into a 2 mM dopamine solution and co-incubated for 10 min. After centrifugation to remove the supernatant, a 1 mg/mL FITC (Sigma, St. Louis, MO, USA) solution was prepared in DMSO. An overnight incubation of the dopamine-treated nanoparticles was conducted with the FITC solution, followed by centrifugation and three washes with distilled water to obtain FITC-labeled cobalt and selenium nanoparticles.

C2C12 cells were cultured in a 24-well plate coated with coverslips and treated with cobalt and selenium nanoparticles for 48 h. The cells were collected, washed with PBS, fixed with PFA for 15 min, and then washed again. Next, the cells were incubated with 0.1% PBST for 15 min and then treated with 5% BSA for 1 h. Finally, the cells were incubated overnight at 4 °C with α-SMA (Abcam, Cambridge, UK, 1:200). The cells were treated with Actin-Tracker Green-488 and Cy3-conjugated goat antirabbit IgG (H + L) for 1 h. Afterward, they were treated with Solarbio’s DAPI for 7 min, followed by a PBS wash. High-quality images were obtained using the Zeiss LSM880 (Zeiss, Oberkochen, Germany).

### 2.5. ROS Detection

C2C12 cells were cultured in a 6-well plate until they reached 70% confluence. They were then treated with different concentrations of Co and Se NPs for 48 h by adding two NPs simultaneously. After treatment, the cells were washed with PBS and incubated for 30 min with a H2DCFDA (2’,7’-dichlorodihydrofluorescein diacetate) reactive oxygen species (ROS) detection kit (Servicebio, Wuhan, China). Fluorescence images were captured using a fluorescence microscope (Leica, Weztlar, Germany). In this study, we assessed the ROS levels in zebrafish after treatment with Co and Se NPs using 500 ng/mL H2DCFDA (2’,7’-dichlorodihydrofluorescein diacetate) staining from Sigma (St. Louis, MO, USA).

### 2.6. Western Blot Analysis

For WB analysis, C2C12 cells were cultured in 6-well plates. The cells were treated with cobalt and selenium nanoparticles for 48 h until 70% fusion was achieved. Cellular proteins were extracted and subjected to electrophoresis using a 4–20% sodium dodecyl sulfate–polyacrylamide gel electrophoresis (SDS-PAGE) system from ACE Biotech (Changzhou, China). The separated proteins were subsequently transferred onto PVDF membranes from Beyotime (Shanghai, China). After blocking with 5% skim milk in 1× TBST, the membranes were incubated overnight at 4 °C with caspase-3 (CST, 1:1000) and cleaved caspase-3 (CST, 1:1000). After incubating with the primary antibody, we probed the blots with a horseradish peroxidase-labeled antirabbit immunoglobulin secondary antibody (BIOSS, Beijing, China) at a dilution ratio of 1:5000 for 1 h.

### 2.7. Flow Cytometry of Annexin V-FITC/PI Double Staining

Cells were cultured in six types of plates until they reached 70% confluence. After that, they were exposed to CoNPs and SeNPs simultaneously for 24 h. Following the treatment, both the cell culture medium and trypsin-digested cells were collected, subjected to centrifugation at 1000× *g* for 5 min, and subsequently resuspended. A total of 5 – 10 ∗ 10^3^ resuspended cells were taken, subjected to another round of centrifugation, and then treated with 5 μL of annexin V-FITC. The cell suspension was incubated in the dark for 10 min. Afterward, 10 μL of propidium iodide staining solution was added. Flow cytometry analysis was conducted using CytoFLEX (ThermoFisher, Waltham, MA, USA).

### 2.8. Quantitative Real-Time PCR

Cells were cultured on 6-well plates and exposed to CoNPs and SeNPs simultaneously for 24 h. Afterward, the cells were washed twice with PBS, and total RNA was extracted using Trizol reagent (Sigma, St. Louis, MO, USA). The RNA concentration was measured, and cDNA synthesis was performed using the SweScript RT II First Strand cDNA Synthesis Kit (Servicebio, Wuhan, China). The SYBR green system was used to execute quantitative polymerase chain reaction (qPCR), and the results were obtained by analyzing 2-ΔΔCt values. The primer sequences for MyoD, myogenin, and Myf5 can be found in Appendix A.

### 2.9. Transmission Electron Microscope

Zebrafish were cultured in a circulating system at 28 °C with a daily light exposure of 14 h, following established protocols for fertilization. At 1 day postfertilization (dpf), fish eggs were exposed to varying concentrations of CoNPs and SeNPs. At 8 dpf, muscle tissue samples were collected from the zebrafish and examined using a transmission electron microscope (TEM). The remaining zebrafish were cultured, and the solutions of CoNPs and SeNPs were changed daily until 3 months postfertilization (mpf). The measurement tools in the ImageJ software (Version 1.52p) were used to measure the length of individual sarcomeres. One sarcomere length was measured from the center of one Z-line to the center of the next Z-line. The lengths of individual sarcomeres were recorded and analyzed. For the zebrafish experiments, nanoparticles were added at 1 dpf and muscle tissue was sampled for TEM at 8 dpf. Survival and malformation rate observations were performed until 10 dpf.

### 2.10. Histological Analyses

HE staining on muscle tissues was performed in zebrafish specimens at 3 mpf. Zebrafish tissue samples were collected after treating cobalt and selenium nanoparticles for three months. The zebrafish tissue samples were fixed in tissue fixing solution (Servicebio, Wuhan, China) for 24 h. After fixation, the samples underwent a dehydration process and were embedded in paraffin. The paraffin-embedded tissues were then sectioned into 2.5 µm slices using a rotary microtome and subjected to hematoxylin and eosin (HE) staining. The measurement tools in the ImageJ software were used to measure the total fiber area, the space between myofibers, and the number of fibers per 500 µm.

### 2.11. Detection of MDA, SOD, and GSH

According to the manufacturer’s instructions, the levels of MDA (malondialdehyde), GSH (glutathione), and SOD (superoxide dismutase) in C2C12 cells were measured using a GSH ELISA Kit (Elabscience, Wuhan, China), MDA Content Detection Kit (Beyotime, Nantong, China), and SOD Activity Detection Kit (Beyotime, Nantong, China).

### 2.12. Statistical Analysis

The experiments were replicated at least three times independently, and the results are presented as mean ± standard deviation. Statistical analyses were performed using SPSS 25 software (IBM, Chicago, IL, USA). Group comparisons were conducted using one-way analysis of variance (ANOVA) for multiple groups and *t*-tests for pairwise comparisons. Statistical significance was set at *p*-values < 0.05.

## 3. Results

### 3.1. Characterization of CoNPs and SeNPs

The CoNPs and SeNPs were found to be composed exclusively of Co and Se, respectively, as confirmed by EDAX analysis (Figure 1A). SEM analysis revealed that the diameters of both types of nanoparticles ranged from 200 to 459 nm (Figure 1B). To characterize the properties of the nanoparticles in aqueous solution, zeta potential measurements were performed, and the results showed that the surface charge of CoNPs was −1.25 mV (Figure 1C), while that of SeNPs was −25.7 mV (Figure 1C). Particle size distribution analysis of CoNPs and SeNPs showed nanomaterials with diameters in the range of 255–459 nm (Figure 1D).

### 3.2. Low-Level SeNPs Inhibited the Cytotoxic Effect of High-Dose CoNPs

To confirm the presence of CoNPs and SeNPs inside the cells, we observed the fluorescence signals of FITC-conjugated CoNPs and SeNPs by confocal microscopy after 2 h treatment with the nanoparticles (Figure 2A). C2C12 cells were treated with concentrations ranging from 0 to 80 μg/mL of CoNPs or 0 to 40 μg/mL of SeNPs. Cell viability gradually decreased with increasing concentrations of CoNPs, reaching 69.6% at 20 μg/mL (Figure 2B). Notably, cell viability increased to 124.7% in the 5 μg/mL SeNP treatment group (Figure 2C). To determine whether SeNPs could inhibit the cytotoxicity caused by CoNPs, we added 5 μg/mL SeNPs to the 20 μg/mL CoNP group after 12 h of treatment and incubated the cells for another 12 h. We compared cell survival in the treatment group receiving CoNPs alone with the group receiving a mixture of CoNPs and SeNPs. The cell survival rate in the CoNP group was still 61.9%, while the survival rate in the mixed group was 84.3% (Figure 2D). The results confirmed that a low concentration of SeNPs was able to inhibit the toxic effects of a high concentration of CoNPs on the cells.

### 3.3. Low-Dose SeNPs Inhibited CoNP-Induced Oxidative Stress in Muscle Cells

H2DCFDA staining revealed that both 20 μg/mL CoNPs and 500 μM H2O2 treatments significantly increased reactive oxygen species (ROS) levels in C2C12 cells compared with the control group, upregulating 10.51 ± 2.82-fold and 15.26 ± 2.11-fold, respectively. In contrast, the ROS levels in the 5 μg/mL SeNP-treated group did not significantly change. The combined treatment with CoNPs and SeNPs significantly decreased the ROS level to 45.7% of that in the 20 μg/mL CoNPs alone treatment group (Figure 3A,B). Additionally, SeNPs were found to decrease ROS production induced by 500 μM H_2_O_2_ by 43.3%. The results indicate that SeNPs have a protective effect against oxidative stress (Figure 3A,B). The MDA results indicate that the addition of 20 μg/mL CoNPs increased the MDA level to 5.50 ± 0.55 nmol/mg compared with the control (2.01 ± 0.11 nmol/mg). However, the addition of 5 μg/mL SeNPs decreased the MDA level to 3.20 ± 0.40 nmol/mg, which was only 58.2% of that of the CoNPs alone treatment (Figure 3C). The GSH assay results indicate that exposure to 20 μg/mL CoNPs reduced the cellular GSH level to 45.7% (7.82 ± 1.50 nmol/mg) of the control group (17.10 ± 1.65 nmol/mg). However, coculturing CoNPs and SeNPs restored the GSH level to 15.82 nmol/mg, which is similar to the control group (Figure 3D). SeNPs did not significantly alter the CoNP-induced SOD downregulation (Figure 3E), but they partially restored the H2O2-induced SOD reduction (4.12 ± 1.26 vs. 6.36 ± 1.71 U/mg). These results suggest that SeNPs can attenuate the oxidative damage induced by CoNPs.

### 3.4. Low-Dose SeNPs Inhibited CoNP-Induced Apoptosis

To investigate the mechanism by which SeNPs reversed the cytotoxicity of CoNPs, we quantitatively detected the expression level of the apoptotic protein caspase-3. It was observed that SeNPs significantly inhibited the increase in cleaved caspase-3 expression induced by CoNPs (Figure 4A,B). The annexin V-FITC/PI double-staining flow cytometry assay showed that the proportion of apoptotic cells in the CoNP-treated group (20.96± 4.16%) was significantly higher than that in the control group (7.31 ± 2.86%). However, the combined treatment with SeNPs and CoNPs resulted in a significant decrease in the apoptotic percentage (12.85 ± 3.80%), although it was still higher than that in the control group and the group treated with SeNPs alone (6.84 ± 2.58%) (Figure 4C,D). The study found that treatment with CoNPs at 20 μg/mL significantly increased apoptosis, while treatment with SeNPs at 5 μg/mL was effective in rescuing the increase in apoptosis induced by CoNPs.

### 3.5. SeNPs Promote the Expression of Myogenic Markers and Protect against Muscle Damage Induced by CoNPs

To investigate the protective effect of SeNPs on CoNP-induced muscle damage, we observed a significant decrease in the fluorescence intensity of α-smooth muscle actin (α-SMA), a myogenic marker, to 39.9% in CoNP-treated C2C12 cells compared with the control group using confocal microscopy. In contrast, the amount of α-SMA fluorescence increased 1.93-fold after combined treatment with SeNPs and CoNPs. This was only slightly lower than the fluorescence intensity of the normal control group. In addition, treatment with SeNPs alone significantly promoted the expression of α-SMA (Figure 5A,B). These results suggest that SeNPs have a protective effect against CoNP-induced muscle damage by promoting the expression of α-SMA. Meanwhile, the qPCR analysis revealed a significant reduction in the expression of myogenic regulatory factors (myogenin, MyoD, and Myf5) in the CoNP-treated group, while SeNP treatment led to a significant increase in these indicators (Figure 5C–E). The treatment involving a mixture of SeNPs and CoNPs resulted in higher expression of myogenesis indicators myogenin and MyoD compared with the control group (Figure 5C,D). The expression of Myf5 was similar to that of the control group (Figure 5E). These results indicate that SeNPs helped to alleviate muscle cytotoxicity induced by CoNPs by promoting the expression of myogenic genes.

### 3.6. Protective Effect of SeNPs against CoNP-Induced Toxicity in Zebrafish

The results of the in vitro study suggest that CoNPs may cause damage to muscle cells, while SeNPs may alleviate the damage caused by CoNPs. To verify these findings, Co and Se NPs were used to treat zebrafish and their effects were observed at 5 dpf. The locomotion trajectories of zebrafish showed a significant decrease in locomotion speed and swimming distance after treatment with 40 μg/mL CoNPs compared with the control group. However, the locomotor performance was similar to that of the control group without significant changes after treatment with 10 μg/mL SeNPs. After treatment with SeNPs and CoNPs, the locomotor ability of zebrafish increased relative to the CoNP-treated group (Figure 6A–C). However, it was still lower than that of the control group. Additionally, the ROS level in 5 dpf zebrafish was significantly higher after treatment with 40 μg/mL CoNPs, about 4.6 ± 0.36 times higher than that of the control group. Following treatment with SeNPs, the ROS level remained relatively stable at 0.90 ± 0.09 when compared with the control group. However, when CoNPs were mixed with SeNPs, the ROS level decreased compared with treatment with CoNPs alone. Specifically, the ROS level was 2.95 ± 0.36-fold of the control (Figure 6D,E), which is consistent with the results of the in vitro experiments.

It was observed that the malformation rate was approximately 13.7% for treatment with CoNPs alone. However, when SeNPs were mixed with CoNPs, the malformation rate decreased to about 6.8%. There was no significant difference between the malformation rate with SeNPs alone treatment and that of the control group (Figure 6F). In addition, after 10 days of treatment, the survival rate of zebrafish in the group treated with CoNPs was 66%, while the survival rate of zebrafish in the group treated with SeNPs was 93%, which was not significantly different from the control group. The survival rate of zebrafish treated with a mixture of Co and Se NPs was 75% (Figure 6G). Taken together, these findings indicate that SeNPs effectively alleviate zebrafish toxicity induced by CoNPs.

### 3.7. Protective Effect of SeNPs against CoNP-Induced Muscle Damage in Zebrafish

TEM observation of 8 dpf zebrafish muscle tissue revealed the successful penetration of Co and Se NPs into the muscle (Figure 7A). This study found that CoNP treatment resulted in muscle segment shortening and pathological changes, including disordered muscle fiber alignment, myogenic fiber breakage, and fibrous tissue proliferation. However, the injury was significantly reduced when SeNPs were administered in combination with CoNPs (Figure 7A,C). It is important to note that SeNP treatment alone had no significant effect on myofiber structure. When 4-month-old zebrafish were treated, we observed by HE staining that CoNPs caused damage to muscle tissue, as evidenced by the reduction of muscle bundles, but SeNPs partially alleviated the muscle tissue damage caused by CoNPs (Figure 7B). The quantitative results showed that the muscle area, the space between muscle fibers, and the density of muscle fibers showed changes consistent with the TEM observations (Figure 7D–F). Taken together, the results indicate that SeNPs have a significant protective effect against CoNP-induced muscle damage in zebrafish.

## 4. Discussion

Cobalt is a widely used component in industry and daily life, and is a major component of biomedical implants, particularly in orthopedic applications [1]. Recent studies have revealed potential nanoparticle release upon implantation, which can lead to the accumulation and migration of particles, resulting in adverse effects such as renal fibrosis and neural damage [3,30]. Skeletal muscle, one of the primary systems in contact with cobalt implants, has been understudied regarding prolonged exposure to CoNPs. This investigation confirms the harmful impact of CoNPs on muscle cells, inducing oxidative stress and apoptosis. In vivo experiments further demonstrate their damaging effects on zebrafish muscle tissue. Previous research indicates that SeNPs have antioxidative and antiapoptotic effects in oxidative stress models [22,28,31]. This study shows that SeNPs effectively counteract the toxic effects induced by CoNPs, protecting both muscle cells and zebrafish from injury, and demonstrating potential detoxification properties. Mechanistic insights suggest that this protective effect may be mediated through antioxidative stress and apoptosis inhibition pathways.

Metal nanoparticles can have various shapes, including spheres, cubes, or rods, with at least one dimension measuring 100 nm or less. In this study, although some of the metal nanoparticles are larger than 100 nm, we still consider them to be in the nanoscale domain since some of their dimensions fall within the nanometer range. This can have implications for the toxicity of nanosized metal particles. Previous studies have shown that nanoparticles can easily penetrate cell membranes, enter cells, and trigger intracellular responses [5]. The internalization of nanoparticles into cells is associated with organelle damage, altered gene expression, and oxidative stress. For instance, nanoparticles can cause harm to the mitochondria and endoplasmic reticulum once they enter the cell. This can disrupt cellular energy metabolism, leading to endoplasmic reticulum stress and protein folding reactions. Additionally, nanoparticles can worsen oxidative stress by affecting cellular signaling pathways, modifying transcription factor activity, or interacting with RNA in the cytoplasm [32]. In this study, we used Co and Se NPs ranging from 200 to 459 nm to treat cells. Using confocal microscopy, we observed FITC-labeled nanoparticles entering cells. We also noted alterations in the viability of C2C12 cells following nanoparticle treatment. These findings suggest that nanoparticles may have an impact on muscle cells.

Elevated levels of reactive oxygen species (ROS) can lead to oxidative stress, which can affect the integrity of cell membranes and mitochondrial function. There is a strong correlation between high ROS levels and intracellular skeletal muscle function; high levels of ROS can cause oxidation of proteins, peroxidation of cell membranes’ lipids, and damage to DNA, which are positively associated with muscle injury [33]. Increased concentrations of cobalt have been linked to higher ROS levels, which can cause oxidative stress and cellular damage [11]. Extended or frequent exposure to cobalt can worsen oxidative stress. This is because cells may not be able to effectively remove free radicals under sustained exposure, leading to a cumulative effect of oxidative stress. CoNPs can also cause cells to produce more inflammatory mediators, such as TNF-α and IL-1β, which can trigger an inflammatory response and further increase oxidative stress [34]. Selenium acts as an antioxidant and enhances the activity of intracellular antioxidative defense systems, effectively clearing ROS to protect cells from oxidative stress-induced damage [35]. Furthermore, SeNPs have been found to regulate the function of various antioxidant enzymes, including superoxide dismutase (SOD). By enhancing the activity of these enzymes, SeNPs help to maintain intracellular redox homeostasis and reduce the production of ROS [36]. In our study, exposure of C2C12 cells to high concentrations of CoNPs resulted in a significant increase in ROS levels. However, exposure to lower concentrations of SeNPs led to a marked reduction in ROS levels. Low concentrations of SeNPs were found to inhibit the elevated ROS levels induced by CoNPs. These findings suggest that SeNPs may counteract the muscular cell damage induced by CoNPs.

Muscle tissue injury involves multiple pathways and mechanisms, including apoptosis, a crucial programmatic cell death process that clears damaged cells and maintains tissue homeostasis [37]. Our study observed a significant upregulation in the expression levels of apoptosis-related genes following CoNP treatment, indicating that tissue cell damage may trigger apoptosis. SeNPs were found to rescue CoNP-induced apoptosis in muscle cells. The relationship between myogenic gene expression and muscle injury is complex. Specific myogenic genes, such as myogenin, MyoD, and Myf5, play key regulatory roles in the repair and regeneration of muscle cells during the injury recovery process [38]. Downregulation of these myogenic genes was observed following CoNP treatment. However, the expression of myogenic genes increased after treatment with SeNPs, effectively mitigating the decline in myogenic gene expression induced by CoNPs. Furthermore, confocal microscopy revealed that SeNP treatment could rescue the decline in α-SMA expression induced by CoNPs. The results indicate that SeNPs enhance myogenic gene expression and prevent apoptosis, thus ameliorating muscle injury caused by CoNPs.

Studies have shown that contact with CoNPs raises oxidative stress levels in rat muscle tissue, resulting in muscle tissue damage [39]. In zebrafish embryos, exposure to high concentrations of CoNPs is toxic and negatively impacts development [40]. Additional studies indicate that SeNPs have the potential to enhance fish growth through antioxidant mechanisms and liver health improvement [35,41]. To test the hypothesis that SeNPs can alleviate muscle tissue damage caused by CoNPs, we created a zebrafish model exposed to both Co and Se NPs. The results indicate that zebrafish treated with CoNPs experienced reduced muscle bundles and significant injury, while those treated with SeNPs showed a notable reduction in muscle injury. Behavioral outcomes support these findings, showing a significant decline in zebrafish locomotor abilities after CoNP treatment, which SeNP treatment effectively mitigated. These findings highlight the potential of SeNPs in mitigating muscle tissue injury caused by CoNPs, including treatment as a top coat or as a complementary medicine treatment. However, this study has limitations as we did not investigate the impact of Co and Se NPs on other organs, which requires further research. The size and shape of the nanoparticles selected may vary from those found in human tissues.

## 5. Conclusions

In summary, our study found that CoNPs induce oxidative stress and apoptosis, which lead to a decrease in the expression of myogenic genes and subsequent muscle tissue damage. However, SeNPs can mitigate the adverse effects of CoNPs by suppressing oxidative stress and apoptosis while promoting the expression of myogenic genes. This highlights the potential of SeNPs to alleviate the detrimental impacts on muscle tissue caused by CoNPs. The application of SeNP coatings has potential as a strategy to counteract the toxicity associated with cobalt alloy in implants, thereby enhancing the safety profile of metallic implants. This approach shows promise for improving the safety of implants.

## Figures and Tables

**Figure 1 toxics-12-00130-f001:**
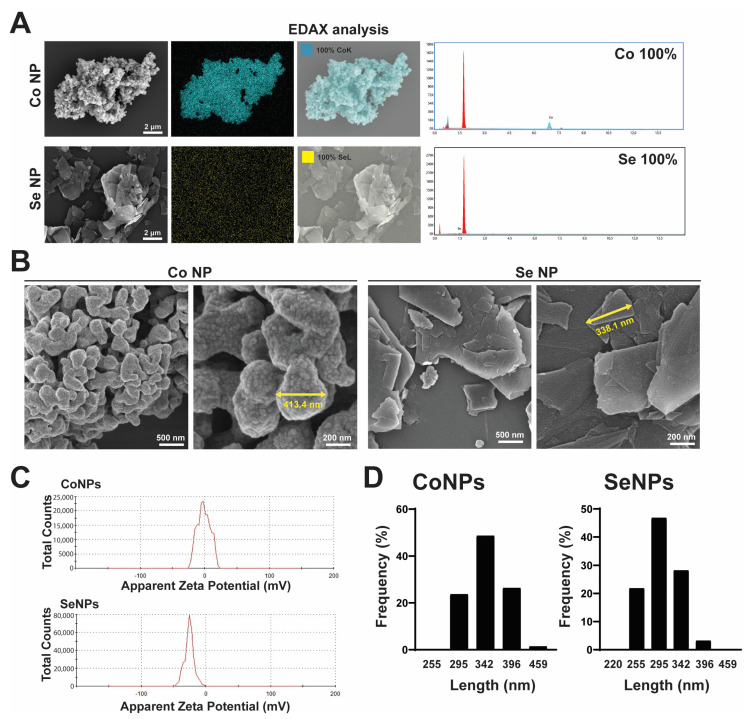
Characteristics of cobalt and selenium nanoparticles. (**A**) EDAX analysis of CoNPs and SeNPs. (**B**) Scanning electron microscopy results of two nanomaterials, SeNP and CoNP, showing nanomaterials with diameters in the range of 200–400 nm. (**C**) ζ-potentials of SeNPs and CoNPs, confirming that SeNPs and CoNPs are negatively charged (surface charges are −25.7 mV and −1.25 mV, respectively). (**D**) Particle size distribution of CoNPs and SeNPs, showing nanomaterials with diameters in the range of 255–459 nm.

**Figure 2 toxics-12-00130-f002:**
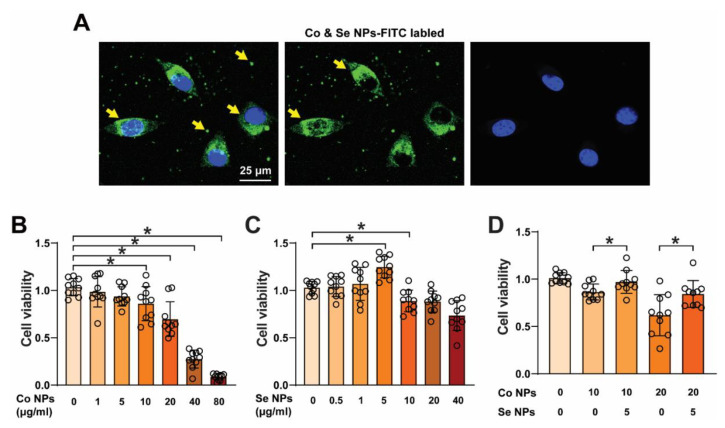
Cellular toxicity induced by cobalt and selenium nanoparticles. (**A**) Cells were treated with 5 μg/mL FITC-labeled CoNPs and SeNPs for 2 h. Confocal microscopy revealed fluorescent signals into the cells. Yellow arrows indicating cells that are filled with green fluorescence, while the culture medium contains free FITC-labeled CoNPs and SeNPs. (**B**–**D**) Viability of C2C12 cells treated with 5–80 μg/mL CoNPs, SeNPs, and mixing of CoNPs and SeNPs were determined by CCK-8. Data are presented as mean ± standard deviation of three identical experiments conducted in triplicate. * Statistically significant difference compared with the controls (*p* < 0.05 for each).

**Figure 3 toxics-12-00130-f003:**
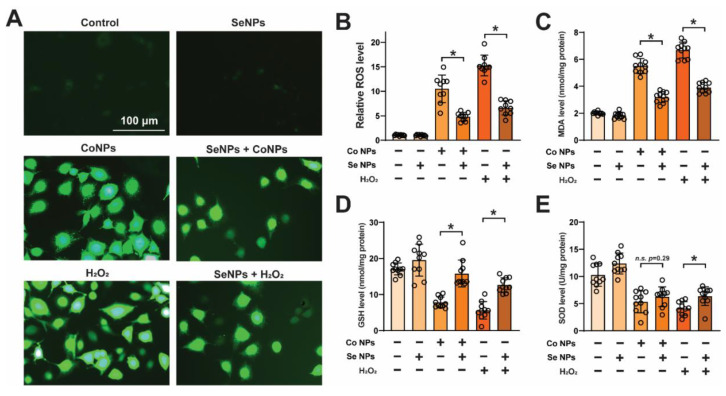
Protective effects of low-dose SeNPs on CoNP-induced oxidative stress in muscle cells. (**A**) C2C12 cells were exposed to control, 20 μg/mL CoNPs, 500 µM H2O2, 5 μg/mL SeNPs with or without H2O2, and mixing of Co and Se NPs. Oxidative stress condition induced by H2O2 treatment for 4 h. Intracellular ROS production was analyzed by H2DCFDA staining. (**B**–**E**) MDA, GSH, and SOD levels were determined in C2C12 cells from control, 20 μg/mL CoNPs, 500 µM H2O2, 5 μg/mL SeNPs with or without H2O2, and mixing of Co and Se NP-treated groups. Data are displayed as mean ± SD; * means *p* < 0.05 between two indicated groups (n = 10).

**Figure 4 toxics-12-00130-f004:**
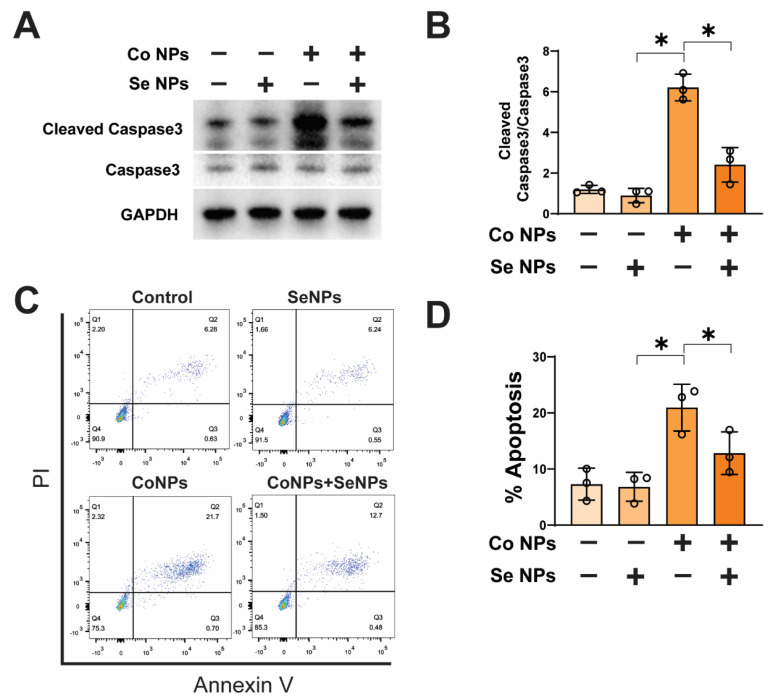
CoNP-induced apoptosis was inhibited by SeNPs in muscle cells. (**A**,**B**) The expression of caspase–3 and cleaved caspase–3 in C2C12 cells treated with 5 μg/mL SeNPs or 20 μg/mL CoNPs with or without SeNPs was analyzed by Western blotting. GAPDH was used as internal reference. Relative expression levels are shown in the graph. (**C**,**D**) The degree of apoptosis of cells subjected to different treatments was determined by annexin V-FITC/PI double-staining flow cytometry. The percentage of apoptotic cells was counted and compared between the different groups. Data are displayed as mean ± SD; * means *p* < 0.05 between two indicated groups (n = 3).

**Figure 5 toxics-12-00130-f005:**
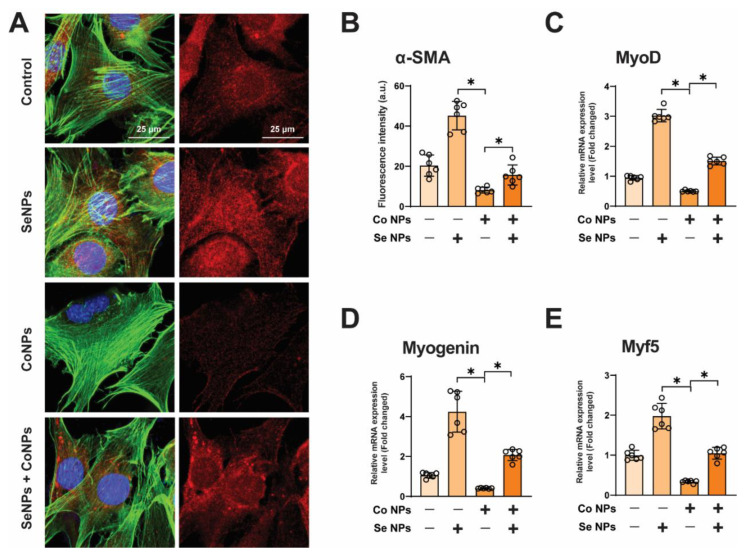
Promotive effects of SeNPs on CoNP-induced inhibition of myogenic differentiation. (**A**) Immunofluorescence staining in SeNPs, CoNPs, and Co Se NP-treated C2C12 cells was analyzed at 4 days of differentiation to detect the effect of different nanoparticles on myogenic differentiation. Nuclei (DAPI, blue), F-actin (Green), and α-SMA (Red) were labeled; scale bar = 25 µm. (**B**) Quantification of α-SMA fluorescence intensity in (**A**) was performed. Data are displayed as mean ± SD. * means *p* < 0.05 between two indicated groups (n = 6). (**C**–**E**) mRNA expression of myogenic markers, MyoD, myogenin, and Myf5, in C2C12 cells treated with 5 μg/mL SeNPs, 20 μg/mL CoNPs with or without SeNPs was analyzed by qPCR. The control group was set to 1.0. Data are displayed as mean ± SD. * means *p* < 0.05 between two indicated groups (n = 6).

**Figure 6 toxics-12-00130-f006:**
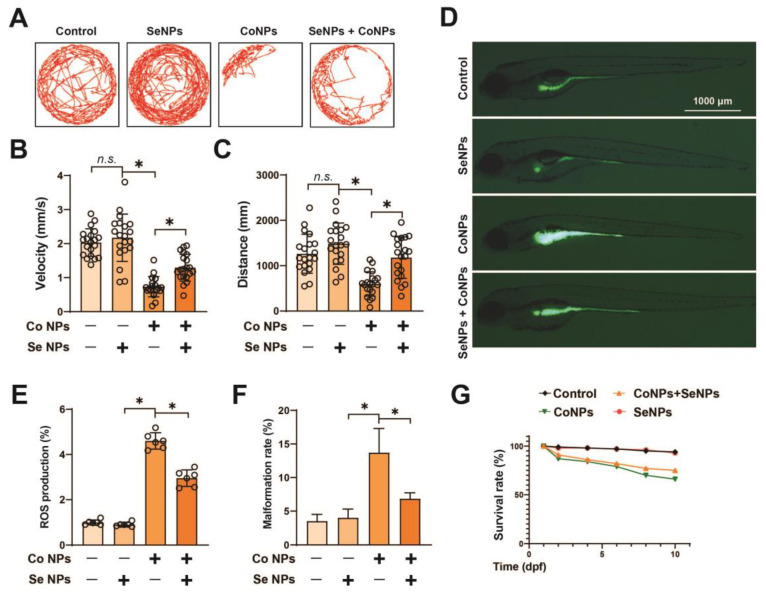
Zebrafish toxicity of cobalt and selenium nanoparticles. (**A**) Locomotion tracks, (**B**) average speed, and (**C**) total movement distance of zebrafish larvae exposed to 40 μg/mL CoNPs, 10 μg/mL SeNPs, and mixing of 40 μg/mL CoNPs and 10 μg/mL SeNPs at 5 dpf (n = 20). (**D**,**E**) ROS production of zebrafish embryos (5 dpf) from control, 40 μg/mL CoNP-, 10 μg/mL SeNP-, and Co Se NP-treated groups was detected and quantified by staining with DCF-DA; * means *p* < 0.05 between two indicated groups (n = 6). (**F**,**G**) Malformation and survival rate of different concentrations of 40 μg/mL CoNP-, 10 μg/mL SeNP-, and Co Se NP-treated embryos in 10 dpf (n = 100). Data are expressed as the mean ± standard deviation (SD); * *p* < 0.05 versus the control group.

**Figure 7 toxics-12-00130-f007:**
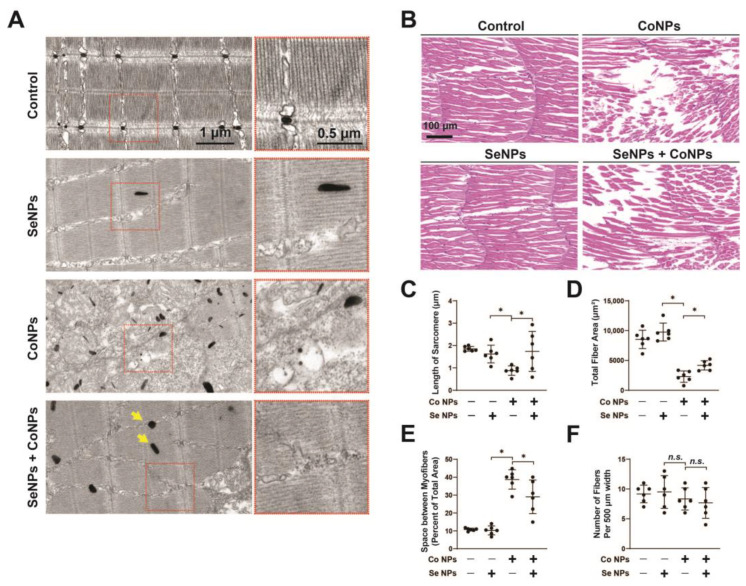
In vivo rescue of muscle toxicity by SeNPs in the presence of CoNPs. (**A**) Zebrafish larvae subject to no treatment (the control), CoNPs, SeNPs, and Co Se NPs for 5 days; muscle samples were examined by TEM and nanoparticle infiltration was visible. Yellow arrows indicate the nanoparticles. Scale bar = 50 µm. (**B**) H&E stain of zebrafish muscle tissue after CoNP, SeNP, and Co Se NP treatment at 7 dpf. Black arrows indicate the damaged muscle fiber. Scale bar = 50 µm. (**C**) Quantification of sarcomere length in TEM results from control, CoNP-, SeNP-, and Co Se NP-treated larvae (n = 6). (**D**–**F**) Quantification of total fiber area, space between myofibers, and number of fibers per 500 µm of zebrafish muscle tissue subject to control, CoNPs, SeNPs, and Co Se NPs for 7 days. Data are displayed as mean ± SD, * means *p* < 0.05 between two indicated groups (n = 6).

## Data Availability

Data is available upon request from the corresponding author.

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
