# Peer review of "Selenium Nanoparticles Attenuate Cobalt Nanoparticle-Induced Skeletal Muscle Injury: A Study Based on Myoblasts and Zebrafish"

_toxics, 2024, doi:10.3390/toxics12020130_

Round 1

Reviewer 1 Report

Comments and Suggestions for Authors
The manuscript investigates the effect of CoNP and the effects of the combination of Co and Se nanoparticles on muscle cells in vitro and in vivo using the C2C12 mouse myoblast cell and the zebrafish model system. The implications of using NPs for biomedical (and other) purposes need to be clarified in detail, as NPs may induce unexpected effects compared to their classical counterparts. As presented in this manuscript, detailed studies of CoNP and SeNP effects have been performed from the morphological to the molecular level. The question is how realistic is exposure to CoNPs via orthopedic implants and what other routes of exposure are possible for these NPs.

INTRODUCTION

As mentioned in the introduction, Co alloys (mainly chromium-based) are used in orthopedic implants and can lead to the release of wear particles and metal ions from the metal implant, resulting in local and systemic toxicity in patients. On the other hand, Co in the form of nanomaterials (it would need to be specified which form of Co nanoparticles) can also have potentially toxic effects. An explanation of the similarities and differences between Co in alloys and in the form of nanoparticles, both structurally and functionally, is needed (in more detail to explain these relationships to us readers). The biomedical use of CoNP mentions the use of CoNP in catalysis or as an effective carrier for cytotoxic drugs or in diagnostic, therefore it is necessary to further explain the possible use of CoNP in orthopedics.

MATERIAL AND METHODS and RESULTS

2.1. Although the nanoparticles are commercially available, it would be necessary to state which synthesis methods were used. According to the simple definition, nanoparticles are generally considered to be materials with an external dimension of 100 nanometers or less. So why can metal-nanoparticles fall under this term (and can they? since the dimensions of the used nanoparticles are larger)?

For each experiment, the schedule and duration of exposure to NPs (CoNP and SeNPs) must be specified, in particular whether the combination of NPs was applied at the same time, as in 3.2. for cell viability stated (for example for ROS detection, for C2C12 cells used for Western blot analysis... and especially in the part related to experiments with zebrafish 2.9., 2.10.).

In 2.2. why were these combinations of Co and SeNP concentrations chosen (presented on fig 2D)?

2.5. How was the level of MDA, GSH and SOD determined?

2.9. How were the selected doses for Co- and Se-NP used in the zebrafish experiments, i.e. was a toxicology study performed or were they taken from previous in vitro experiments?

3.2. How did you obtained FITC-conjugated Co and Se-nanoparticles?

3.3. What was concentration of H2DCFDA applied to C2C12 cells and to zebrafish in 3.6.?

3.6. Survival rate groups are not well marked, they do not match the results (Fig 6, graf G).

3.7. Provide bar for Figure 7A inserts, as well as a description of the morphometric measurements of the presented parameters (in Material and methods section).

DISCUSSION

The discussion is largely based on the repetition of results, while the large number of results presented still offers the possibility of a more comprehensive insight into the mechanism of action of NPs, which includes both negative (toxic) effects of CoNps and ameliorative effects of SeNPs.

For example, the results related to oxidative stress (its induction by CoNPs and its attenuation by SeNPs), i.e. the parameters determined, can be discussed and thereby provide a better insight into the processes in muscle cells exposed to NPs.

Reviewer 2 Report

Comments and Suggestions for Authors

The title is short and easy to remember. It is also accurate and concise, while including all the essential information about the study.

Abstract is sufficient as it provides a clear and concise overview of the study's purpose, methods, findings, and conclusions. It is also well-structured and easy to read.

Overall, the choice of statistical analysis seems appropriate for this study. The researchers used appropriate methods to analyze their data and set a reasonable threshold for statistical significance. Histological analyses: the authors should provide a more detailed description of the protocol.

Overall, the study is well-designed, well-conducted, and well-written.

Questions:

Are there studies on the prolonged impact of CoNPs on skeletal muscles, especially in the context of orthopedic implants?

Do the studies provide detailed information on the protective mechanism of SeNPs against the toxic effects of CoNPs on muscle cells?

What are the consequences of nanoparticles penetrating into cells, especially in terms of organelle damage, changes in gene expression, and oxidative stress?

Is there a clear correlation between high ROS levels and skeletal muscle function, especially in the context of muscle injuries?

Do the results suggest potential applications of SeNPs in treating or preventing muscle damage caused by CoNPs?

What are the specific changes in the expression of genes related to apoptosis and myogenesis after exposure to CoNPs, and how do SeNPs influence these processes?

Do experiments on the zebrafish model provide an adequate analogy to the effects of CoNPs on muscles in humans?

Does the study suggest potential clinical applications of SeNPs in reducing the toxicity associated with cobalt-containing implants?

Does the study suggest the need for further research, especially regarding the impact of CoNPs and SeNPs on other organs and differences in the size and shape of nanoparticles compared to those present in human tissues?

Round 2

Reviewer 1 Report

Comments and Suggestions for Authors

I thank the authors for their responses and find that the Introduction and Discussion sections have been improved with more explanation and information. In the interest of future readers of this study, some explanations should be added. I apologize in advance if I overlooked it, but I did not find explanations in the text of the manuscript regarding the chosen concentrations of nanoparticles as well as the duration of exposure to CoNPs, SeNPs as well as the duration of exposure to CoNPs plus SeNPs and the time schedule of exposure. In more detail:

Comments 4 and Response 4: The method for obtaining the nanoparticles used should be included in the text of the manuscript in section 2.1. A further explanation about the size of the metal nanoparticles used, why metal nanoparticles of the above sizes can be considered nanoparticles should be included in the text, preferably a few sentences in the Discussion, first or second paragraph, whichever you think is better (authors provide explanations in Response 2). 

Comments 5 and Response 5: The duration of exposure to nanoparticles should be
included in the text as important information (as already written in Comment 5).
The data related to duration of exposure to nanoparticles in the sets of in vitro
experiments and in the experiments with zebrafish are mainly omitted.
At the same time, it would be logical that the duration of exposure to nanoparticles
is not different in all experiments related to the in vitro approach, as this parameter
would not be another variable in the experiments; in 2.3. section (in Ln 135) add the
time of exposure to nanoparticles; also add an explanation why the indicated
concentrations of nanoparticles were selected for further experiments
(in answer 6 the explanation is provided); in 2.5. add time of exposure to nanoparticles
(in Ln 163); in 2.6. add the same; in 2.7. it was stated that the exposure to
nanoparticles lasted 24 hours (Ln 185); same for 2.8. – 24 hours exposure time
(so the times are listed here); it should also be explained how the combination of
nanoparticles was used, were the cells exposed to the mixture of nanoparticles at the
same time or were the SeNPs added to the CoNPs after a certain time?
Regarding exposure to nanoparticles in experiments with zebrafish, it was stated
that the Co and/or SeNP treatment lasted 8 days (Ln 206) for EM samples,
while in Ln 215 it is mentioned 3 months of exposure to nanoparticles, while
survival rate treatment lasted 10 days (Ln 369), while Ln 409 mentions 5
days of exposure to nanoparticles... etc.? So, this should be clarified in the
Material and Methods section.

Comments 11. The corrections were not made for Figure 6, graph G. According to the text in the results, the control and SeNP groups have a survival rate of over 90%, but according to the graph, the SeNP group (green line) has the lowest survival rate!

Comments 12. Provide the bar for Figure 7A inserts!

Reviewer 2 Report

Comments and Suggestions for Authors

Accept

Round 3

Reviewer 1 Report

Comments and Suggestions for Authors

The manuscript is ready for acceptance and publication in its present form.